# Animal Protein Sources as a Substitute for Fishmeal in Aquaculture Diets: A Systematic Review and Meta-Analysis

**Rendani Luthada-Raswiswi** [1,*] 📷, **Samson Mukaratirwa** [2,3,*] **and Gordon O'Brien** [1,4]

1 School of Life Sciences, College of Agriculture, Engineering and Science, Pietermaritzburg, University of KwaZulu Natal, Scottsville 3209, South Africa; Gordon.Obrien@ump.ac.za
2 School of Life Sciences, College of Agriculture, Engineering and Science, Westville Campus, University of KwaZulu Natal, Durban 4001, South Africa
3 One Health Center for Zoonoses and Tropical and Veterinary Medicine, School of Veterinary Medicine, Ross University, Basseterre KN0101, St Kitts
4 School of Biology and Environmental Sciences, Mpumalanga University, Mbombela 1200, South Africa
* Correspondence: Luthada-RaswiswiR@ukzn.ac.za (R.L.-R.); SMukaratirwa@rossvet.edu.kn (S.M.)

**Abstract:** Fishmeal is the main source of dietary protein for most commercially farmed fish species. However, fishmeal prices have been raised even further because of competition with domestic animals, shortage in world fishmeal supply, and increased demand. Increased fishmeal prices have contributed to the quest for alternatives necessary to replace fishmeal as a global research priority. A literature search was conducted using these terms on Google Scholar and EBSCOhost; fishmeal replacement in fish feeds, fishmeal alternatives in fish feeds, animal protein sources in aquaculture, insects in fish feeds, terrestrial by-products, and fishery by-products. To calculate the variation between experiments, a random effect model was used. Results indicated that different fish species, sizes, and inclusion levels were used in the various studies and showed that the use of insects, terrestrial by-products, and fishery by-products has some limitations. Despite these drawbacks, the use of animal protein sources as a replacement for fishmeal in fish diets has had a positive impact on the feed conversion ratio, variable growth rate, final weight, and survival rate of different types of fish species of different size groups. Findings also showed that some animal by-products had not been assessed as a protein source in aquaculture or animal feeds, and future studies are recommended.

**Keywords:** aquaculture; animal protein sources; fish; fishmeal; feeds

## 1. Introduction

In terms of species cultured and production systems used, aquaculture is a diverse industry [1]. According to [2], by producing fish with minimal environmental impact and maximum benefit for society, aquaculture is predicted to contribute more effectively to economic development, international food safety, nutritional well-being, and poverty reduction. Regardless of the cultivated systems within which fish are grown and species involved, production, growth, and health of fish depend totally on a supply of adequate nutrients both in quantity and quality [3]. The quality of the protein ingredient used in feed formulation is generally known to have effects on the nutritional value of fish diets produced [4]. According to [5], aquaculture production (66 million tons) exceeded global beef production (63 million tons) for the first time in 2012. Increased aquaculture production means that more than half of the fish being consumed by humans worldwide is produced by aquaculture [6]. The demand for feed resources, particularly for prime quality protein fishmeal, has increased because of the global supply of fish as aquaculture production increases [3].

For both carnivorous and omnivorous species used in aquaculture, fishmeal has been used as an essential protein source, and many aquaculture formulations/feeds have a higher percentage of fishmeal than feeds of other animal species [7]. Fluctuations in supply,

price, and quality of fishmeal present considerable risks because fishmeal is dependable solely on an ingredient by people. Therefore, the identification, development, and utilization of alternatives to fishmeal in diets in aquaculture remain a high priority as a risk reduction strategy [8]. Competitive price, full availability, ease of handling, shipping, storage, and use in feed production are features that a candidate ingredient must possess to be a viable alternative feedstuff to fishmeal in aquaculture feeds [9]. Additionally, it should have high protein content, favorable amino acid profile, high nutrient digestibility, low fiber levels, starch, non-soluble carbohydrates, which are nutritional characteristics [9].

The more expensive fishmeal has been replaced by several sources of plant protein, single-cell protein, and animal protein in part or in full [10]. Due to higher protein and lipid content, superior essential amino acids, and excellent palatability, animal protein sources have commonly been considered ideal substitute protein sources to replace fishmeal in formulating fish diets [11,12]. According to [13], animal-derived protein demand is expected to double by 2050 globally. Furthermore, future needs for both food and feed are expected to grow by 70%. According to [14], to provide the mandatory quantities of high-quality protein to fulfill the increasing demand, new initiatives are needed. Several animal protein sources from insects, land by-products and fisheries by-products have been evaluated as possible feed ingredients in fish production [15–19]. However, no documented studies comparing animal protein sources in diet and control diet. The purpose of this study was to conduct a systematic review of published articles on animal protein sources used in aquaculture and assess the results of recommended diets against the control diet.

## 2. Materials and Methods

A systematic search of published literature on Google Scholar and EBSCOhost from 1999 to 2019 was carried out using the following terms or phrases: Fishmeal replacements in fish feeds, fishmeal alternatives in fish diets, animal protein sources in aquaculture, insects in fish feeds, terrestrial by-product, and fishery by-products. By reading through the titles and abstracts, the papers were found and screened. In addition, of the selected articles, the reference and bibliographic lists were screened as potential leads to additional relevant studies for inclusion. In Endnote reference manager version x7.7.1 (Clarivate Analytics, Philadelphia, PA, USA), full-text articles for studies including animal protein sources, were retrieved and managed. An article was included in the review if published between 1999 and 2019 and reported on 3 or all 4 of the following on experimental animals: Specific growth rate, final weight, feed conversion ratio, and survival rate. Studies with less than 4 protein levels tested, and those with no standard error on results were excluded. Furthermore, editorial material, book chapters, and conference papers were excluded. Meta-analysis was conducted for final weight, specific growth rate, feed conversion ratio, and survival rate, separately in a Microsoft Excel Spreadsheet using formulas and procedure described by [20] as follows after entering study Authors and year, events, and sample size for each study included:

1. Calculated the outcome (es) = number of events/the sample size
2. Calculated Standard Error (SE) = Square root of the outcomes/sample size
3. Variance (Var) $= SE^2$
4. Computed the individual study weights (W) $= 1/SE^2$
5. Computed each weighted effect size (W*es) = each effect size multiplied by study weight
6. $W*es^2$ and $W^2$ were calculated.

     All values of each variable were added to have the sum.

7. Calculated Q $= \sum(W*ES^2)-[\sum(W*ES)]^2/\sum W$, Q test to measure heterogeneity among studies.
8. $I^2$ index = (Q-degree of freedom (df)/Q*100, was calculated to quantify heterogeneity, Degree of freedom (df) was calculated as the total number of studies minus 1. If values of $I^2$ index were 0%, ≤25%, 50%, or 75%, the $I^2$ index was interpreted as no, low, moderate, or high heterogeneity, respectively.

9. Decided on the effect summary model. Random Effect Model was used because we assumed that the variability in studies was not due to sampling errors only but also in the population of effects. Furthermore, the Random Effect Model was used to measure the variability between studies, considering that other studies, which were not included in the meta-analysis at hand, could be unpublished, ignored in the systematic literature quest, or to be conducted in the future [21]. The weight of each study was adjusted with a constant $(V) = Q\text{-}df/\sum W\text{-}(\sum W^2/\sum W)$. However, we computed $w^2$ first and then the sum of $w^2$ ($\sum W^2$), which was not computed yet.

10. New weight for each study was calculated using $W_v = 1/(SE^2 + V)$.

11. Weighted effect size (W*es), $W^*es^2$, $Wv^2$, $Q_v$, and $I^2_v$ were computed using the new weight ($W_v$) as in steps 5–8.

12. Calculated the effect summary as $es_v = \sum(Wv^*es)/\sum W_v$ and standard error as $SEes_v = \sqrt{1/\sum W_v}$

13. The lower and upper confidence intervals were calculated as $es_v - (1.96^*SEes_v)$ and $es_v + (1.96^*SEes_v)$, respectively.

14. Figures in results (excluding Figure 1) were drawn using the weights, prevalence, and confidence intervals calculated above.

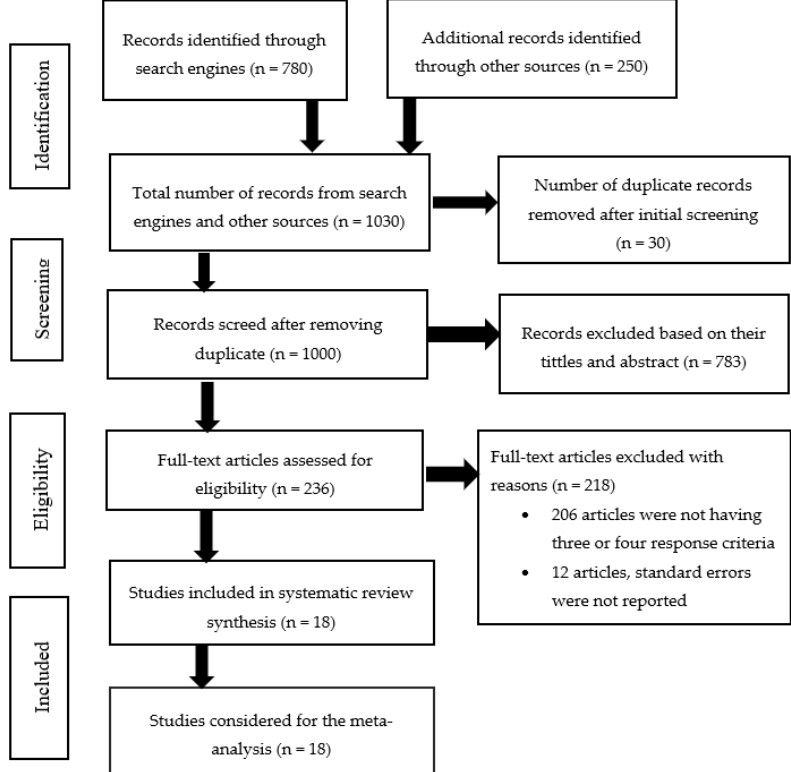

**Figure 1.** Flow chart of the study selection process for systematic review and meta-analysis of animal protein sources as a fishmeal replacement in aquaculture diets.

## 3. Results

There were 1030 articles obtained from search engines, and additional records were identified through other sources. There were 30 articles removed as duplicates after initial screening. Based on their names and abstracts, 783 publications were omitted because they did not follow the requirements of reporting on three or all four of the following on experimental animals: Specific growth rate, final weight, feed conversion ratio, and survival rate, have four or more protein levels tested, and others have no standard error on results. Eligibility was evaluated for 217 articles, and 18 articles were included in the systematic review and meta-analysis (Figure 1).

### 3.1. Fish Species Used and Recommended Levels of Animal Protein Sources

Results from the review articles showed that animal protein sources replacing fishmeal ranged from insects (Mopane worms (*Imbrasia belina*), grasshoppers (*Zonocerus variegatus*), field crickets (*Gryllus bimaculatus*), blowfly maggot (*Chrysomya megacephala*), black soldier fly (*Hermetia illucens*) and superworm (*Zophobas morio*), terrestrial animal by-products (fermented feather meal, feather meal, poultry by-products, meat and bone meal, and blood meal), and fishery by-products (fish silage, shrimp head meal and krill meal) (Table 2). Furthermore, a variety of fish species such as *Oreochromis mossambicus*, *Clarias gariepinus*, *Oreochromis niloticus*, *Sparus aurata*, *Dicentrarchus labrax*, *Scophthamus maeotinus*, *Lutjanus guttatus*, *Ophiocephalus argus*, Red tilapia (*O. mossambicus* × *O. niloticus* × *Oreochromis aureus*), and *Acipenser glueldenstaedtii* (which were not selected but reported because it is important to know when reporting for protein sources used) have been used. Animal protein sources inclusion levels in the diets ranged from 0%, 5%, 10%, 20%, 25%, 30%, 40%, 50%, 60%, 75%, to 100%. Recommended levels of animal protein sources in feeds were 20% for feather and shrimp head meal for *C. gariepinus*, 20% of meat and bone meal for *Op. argus*, 25% of superworm, poultry by-product and grasshopper meal for *L. guttatus* and *C. gariepinus* respectively, 30% of krill meal for *A. glueldenstaedtii*, 20–50% of fermented feather meal for *O. niloticus*, 50% of poultry by-products and fish silage for *O. niloticus* and Red tilapia (*O. mossambicus* × *O. niloticus* × *O. aureus*), respectively, 60% of mopane worm meal for *O. mossambicus* and 100% of field cricket meal for *C. gariepinus*.

**Table 1.** Summary of studies that assessed animal protein sources as a fishmeal replacement in fish diets in aquaculture. Final weight (FW in grams), specific growth rate (SGR in percentage (%)), feed conversion ratio (FCR), and survival rate (SR in %) were used as the assessment parameters to measure response.

| Protein Sources Replacing Fish Meal | Fish Species | Recommended Levels of Feed (%) | Duration of Experiment (Days) | Feeding Frequency (Times/Day) | Initial Weight IW (g) | Outcomes for Recommended Levels | | | | References |
|---|---|---|---|---|---|---|---|---|---|---|
| | | | | | | FW (g) | SGR (%) | FCR | SR (%) | |
| **Insects** | | | | | | | | | | |
| Mopane worm (*Imbrasia belina*) | *Oreochromis mossambicus* | 60 | 51 | 2 | 242.40 | 1221.10 | 3.16 | 1.25 | 100 | [22] |
| Grasshopper (*Zonocerus variegatus*) | *Clarias gariepinus* | 25 | 56 | 2 | 1.32 | 5.75 | 2.64 | 1.51 | 100 | [23] |
| Field Cricket (*Gryllus bimaculatus*) | *Clarias gariepinus* | 100 | 56 | 2 | 4.82 | 19.50 | 2.32 | 2.20 | 93.30 | [24] |
| Blowfly Maggot (*Chrysomya megacephala*) | *Oreochromis sp.* | 100 | 60 | 2 | 3.0 | 10.63 | 2.02 | 1.34 | 80.0 | [25] |
| Black soldier fly (*Hermetia illucens*) | *Salmo salar* | 66 | 112 | 2 | 1386 | 3721 | 0.9 | 1.1 | NR | [26] |
| Superworm (*Zophobas morio*) | *Oreochromis niloticus* | 25 | 56 | 2 | 5.57 | 10.11 | 1.02 | 1.25 | 100 | [27] |
| **Terrestrial animal by-products** | | | | | | | | | | |
| Fermented feather meal | *Oreochromis. niloticus* | 25–50 | 84 | 2 | 122.81 | 222.35 | NR | 1.73 | 100 | [15] |
| Feather meal | *Clarias gariepinus* | 20 | 28 | 2 | 2.85 | NR | 7.89 | 1.34 | 88.89 | [18] |
| Poultry by-products | *Lutjanus guttatus* | 25 | 84 | 3 | 11.0 | 36.17 | 1.43 | 1.20 | 100 | [28] |
| Poultry by-products | *Oreochromis niloticus* | 50 | 84 | NR | 0.88 | 10.19 | 2.70 | 1.40 | 100 | [17] |
| Poultry by-product | *Dicentrarchus labrax* | 60 | 70 | 3 | 0.73 | 8.28 | 3.52 | 2.24 | 94 | [29] |
| Poultry by-product | *Scophthalmus maeoticus* | 25 | 60 | 2 | 18 | 29.38 | 0.18 | 0.91 | 100 | [30] |
| Poultry by-product | *Oreochromis niloticus* | 100 | 120 | 2 | 1.5 | 54.3 | 2.99 | 1.34 | NR | [31] |
| Blood meal | *Clarias gariepinus* | 50 | 86 | 2 | 10.32 | 66.50 | 1.03 | 0.86 | 100 | [32] |
| Meat and bone meal | *Ophiocephalus argus* | 20 | 70 | 3 | 12.11 | 138.67 | 3.48 | 1.24 | 94.2 | [33] |

**Table 2.** Summary of studies that assessed animal protein sources as a fishmeal replacement in fish diets in aquaculture. Final weight (FW in grams), specific growth rate (SGR in percentage (%)), feed conversion ratio (FCR), and survival rate (SR in %) were used as the assessment parameters to measure response.

| Protein Sources Replacing Fish Meal | Fish Species | Recommended Levels of Feed (%) | Duration of Experiment (Days) | Feeding Frequency (Times/Day) | Initial Weight IW (g) | Outcomes for Recommended Levels | | | | References |
|---|---|---|---|---|---|---|---|---|---|---|
| | | | | | | FW (g) | SGR (%) | FCR | SR (%) | |
| | | | | Fishery by-products | | | | | | |
| Fish silage | Red tilapia (*Oreochromis mossambicus* × *Oreochromis niloticus* × *Oreochromis aureus*) | 50 | 84 | NR | 2.18 | 28.05 | 3.04 | 1.35 | NR | [34] |
| Shrimp head meal | *Clarias gariepinus* | 20 | 84 | NR | 12.1 | 32.8 | 1.19 | 2.50 | NR | [35] |
| Krill meal | *Acipenser glueldenstaedtii* | 30 | 200 | NR | 483 | NR | 0.56 | 1.10 | 83 | [36] |

NR = Not recorded.

### 3.2. Values for Final Weight, Specific Growth Rate, Feed Conversion Ratio, and Survival Ratio

Values for final weight, specific growth rate, feed conversion ratio, and survival ratio for recommended levels of animal protein sources in feeds for different fish species are shown in Table 2. Assessment of the initial and final weights for all recommended levels of animal protein sources fed showed weight gain for all fish species involved in the experiments (Table 2). The specific growth rate ranged from 0.56% to 7.89%. Feed conversion ratios of 1.25, 1.51, and 2.20 were reported for O. mossambicus, *C. gariepinus*, and *C. gariepinus*, which were fed insect meal (*I. belina*, *Z. variegatus*, and *G. bimaculatus*), respectively. For terrestrial by-products (fermented feather meal, feather meal, poultry by-products, poultry by-products, and meat and bone meal), feed conversion ratios of 1.73, 1.34, 1.20, 140, and 1.24 were obtained for *O. niloticus*, *C. gariepinus*, *L. guttatus*, *O. niloticus*, and *Op. argus*, respectively. Feed conversion ratios of 1.35, 2.50, and 1.10 were obtained in Red tilapia (*O. mossambicus* × *O. niloticus* × *O. aureus*), *C. gariepinus*, and *A. glueldenstaedtii* fed fishery-by products (fish silage, shrimp head meal, and krill meal), respectively. Survival rate ranged from 83% to 100%, except for Red tilapia (*O. mossambicus* × *O. niloticus* × *O. aureus*) and *C. gariepinus*, which were fed fish silage and shrimp head meal, respectively, where the survival rate was not reported.

### 3.3. Meta-Analysis

For the meta-analysis, data from studies analyzed were grouped into final weight, specific growth rate, feed conversion ratio, and survival rate (Table 3, which summarizes results shown in Figures 2–5). Samples analyzed were 1335, 1430, 1450, and 1307 for final weight, specific growth rate, feed conversion ratio, and survival rate, respectively. Results showed the overall effect size of 9015 (95% confidence interval (CI) 6,110,058.3 to 6,110,177.58), 10 (95% CI 32 to 21), 10 (95% CI 24 to 13), and 546 (95% CI 350 to 572) for final weight, specific growth rate, feed conversion ratio, and survival rate, respectively (Figures 2–5). Effect summary for all Figures 2–5 do not touch or cross the center line, meaning that meta-analysis results indicate a statistically significant difference. The level of heterogeneity observed were $I^2$ = 99.70%, $I^2$ = −17.73%, $I^2$ = −25.79%, and $I^2$ = 101.08% for final weight, specific growth rate, feed conversion ratio, and survival rate, respectively (Table 3).

**Table 3.** Weights, prevalence (95 % CI), effect summary, I² index, and degree of freedom for final weight, specific growth rate, feed conversion rate, and survival rate for different studies included in the meta-analysis.

| | Final Weight | | Specific Growth Rate | | Feed Conversion Ratio | | Survival Rate | |
|---|---|---|---|---|---|---|---|---|
| **Reference** | **Weight** | **Prevalence (95% CI)** | **Weight** | **Prevalence (95% CI)** | **Weight** | **Prevalence (95% CI)** | **Weight** | **Prevalence (95% CI)** |
| [23] | 21 | 47 (31–168) | 50 | 23 (32– 19) | 60 | 17 (26–9) | 1 | 927 (637–839) |
| [15] | 2 | 1041 (779–978) | - | - | 174 | 12 (21–14) | 4 | 500 (301–503) |
| [17] | 50 | 39 (28–170) | 170 | 12 (29–23) | 225 | 9 (22–14) | 4 | 500 (301–503) |
| [18] | - | - | 56 | 27 (26–25) | 101 | 15 (18–18) | 3 | 435 (228–430) |
| [36] | 709 | 141 (61–138) | 1,851,851.8 | 0.054 (26–26) | 1,021,450.5 | 0.1 (24–11) | 11,869.4 | 8 (94–108) |
| [28] | 7 | 228 (2–197) | 166 | 9 (32–20) | 180 | 8 (22–13) | 2 | 663 (433–635) |
| [34] | 13 | 149 (70–129) | 128 | 16 (27–24) | 327 | 6 (20–15) | - | - |
| [35] | 13 | 157 (65–129) | 354 | 6 (30–22) | 153 | 13 (19–17) | - | - |
| [22] | - | - | 3236 | 3 (26–25) | 7299 | 1 (20–15) | 100 | 100 (21–181) |
| [24] | 17 | 88 (70–134) | 138 | 11 (31–20) | 73 | 21 (27–8) | 3 | 604 (380–1085) |
| [33] | 12 | 334 (112–311) | 466 | 9 (26–25) | 1214 | 3 (20–15) | 17 | 231 (83–286) |
| [25] | 9.09 | 106 (76–122) | 49.59 | 20 (33–18) | 74.63 | 13 (27–8) | 1.25 | 800 (524–726) |
| [26] | 0.002 | 124,030 (85,539,850–85,540,050) | 10.01 | 30(58–6) | 8.14 | 37 (50–15) | - | - |
| [27] | 0.01 | 101 (76–122) | 98.03 | 10(35–16) | 78.13 | 13 (27–8) | 1 | 1000 (702–905) |
| [29] | 76.92 | 33 (24–174) | 178.57 | 14 ((26–25) | 277.77 | 9 (20–14) | 6.65 | 376 (199–401) |
| [30] | 7.72 | 196 (31–168) | 1275.51 | 1 (30–22) | 243.9 | 6 (24–11) | 2.27 | 667 (435–637) |
| [31] | 181.82 | 54(49–150) | 3448.28 | 3 (26–25) | 7692.3 | 1 (19–17) | - | - |
| [32] | 2.17 | 554 (269–495) | 140.85 | 9(34–18) | 169.49 | 7 (26–10) | 1.44 | 833 (569–771) |
| **Effect summary** | | 9015 (6,110,058.3–6,110,177.58) | | 9.9 (24–13) | | 10 (32–21) | | 546 (350–572) |
| **Random effect model (I²)** | | 99.40 | | −7.73 | | −27.791 | | 101.08 |
| **Degree of freedom (df)** | | 17 | | 16 | | 17 | | 13 |

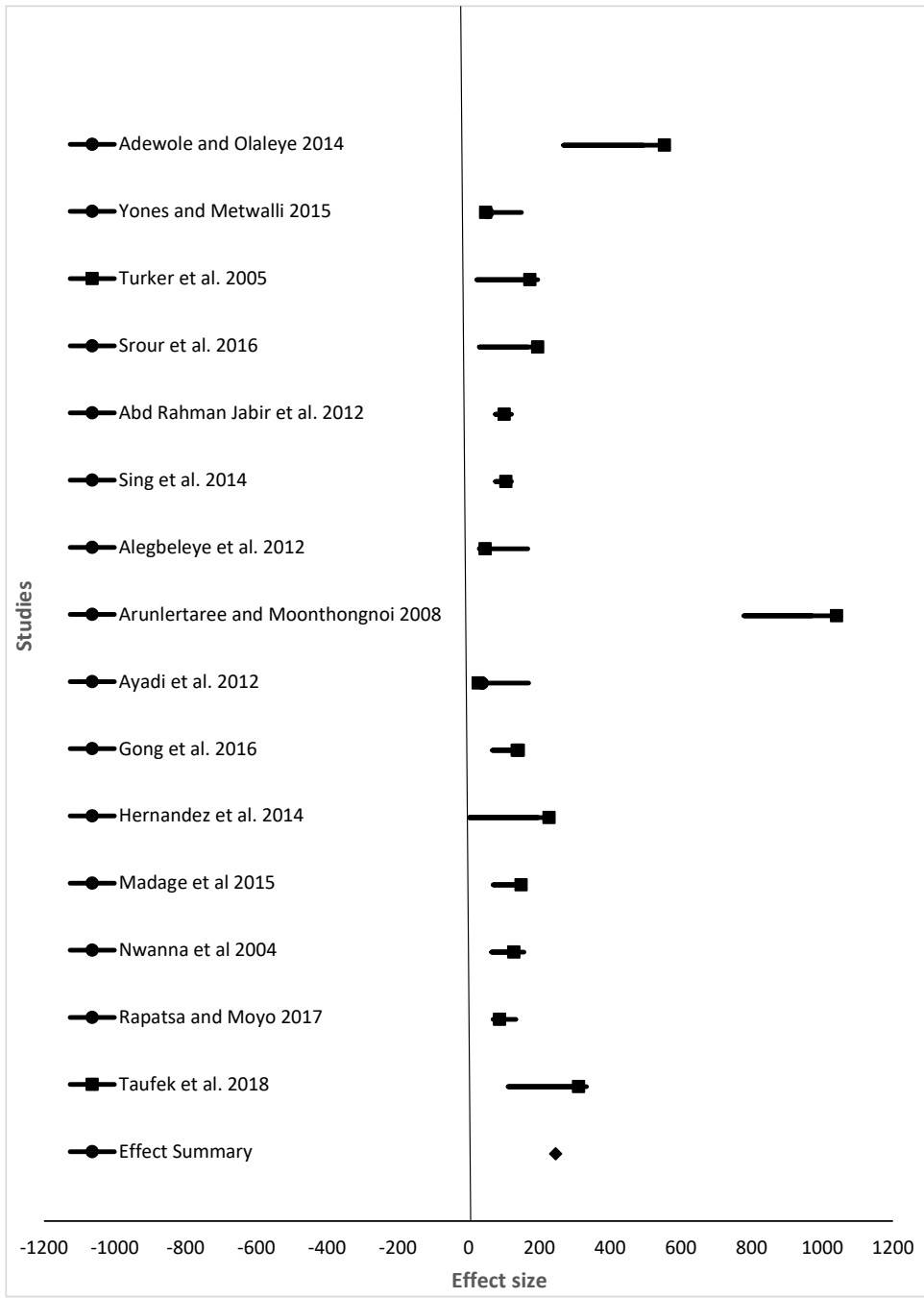

**Figure 2.** The effect size of final weight (%) of fish from different studies fed different animal protein sources compared to fishmeal as a protein source.

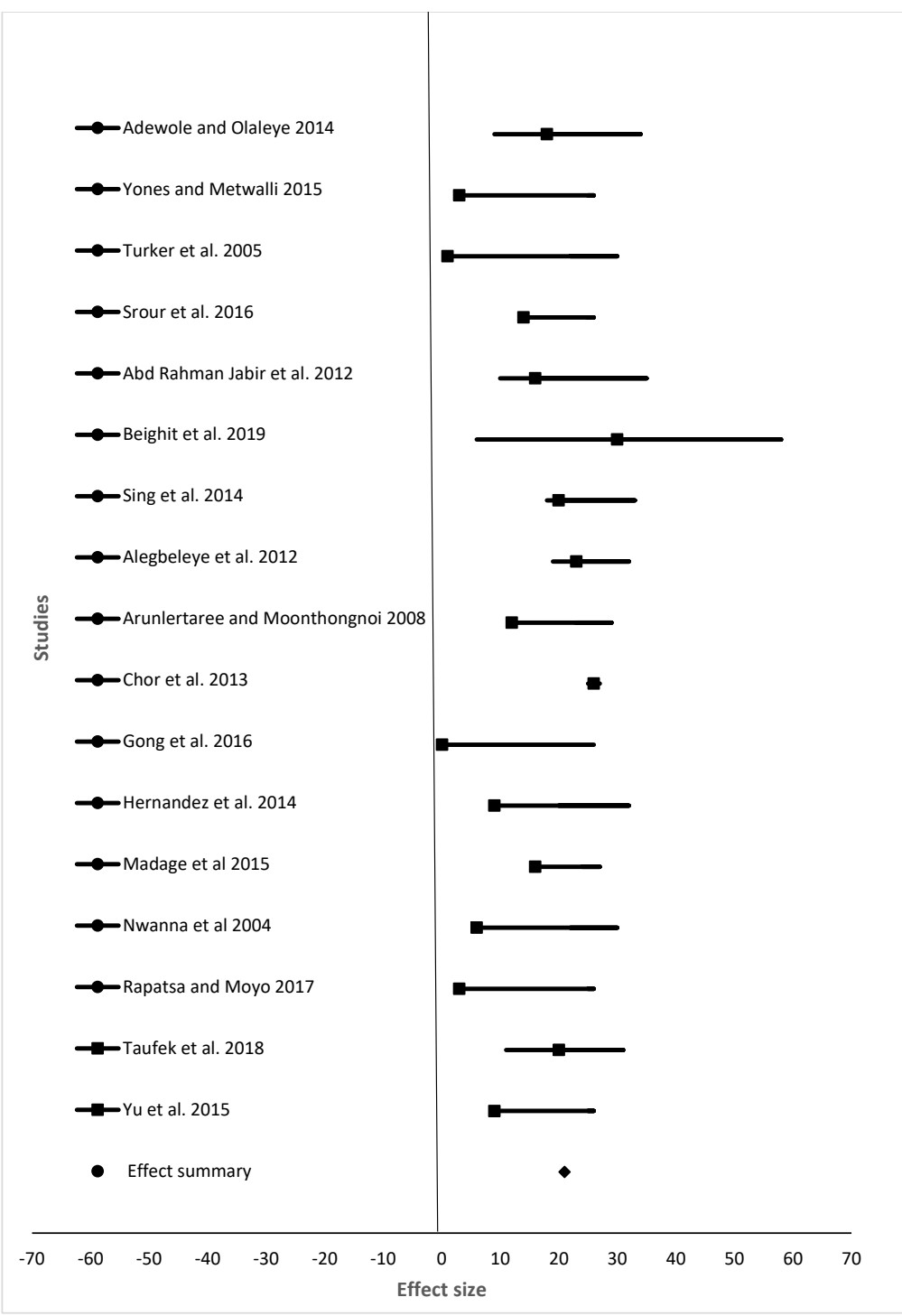

**Figure 3.** The effect size of specific growth rate (%) of fish from different studies fed different animal protein sources compared to fishmeal as a protein source.

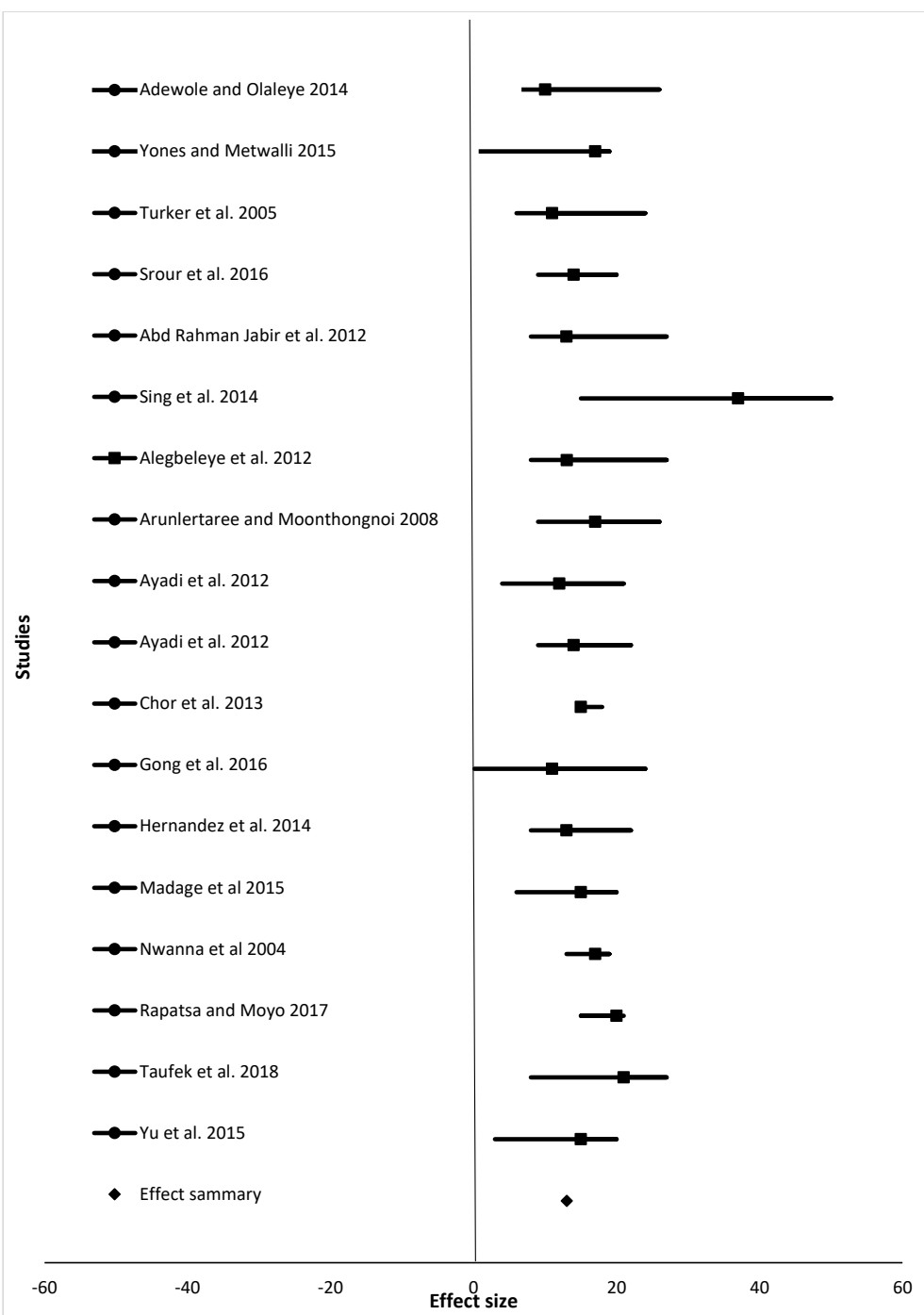

**Figure 4.** The effect size of feed conversion ratio (%) of fish from different studies fed different animal protein sources compared to fishmeal as a protein source.

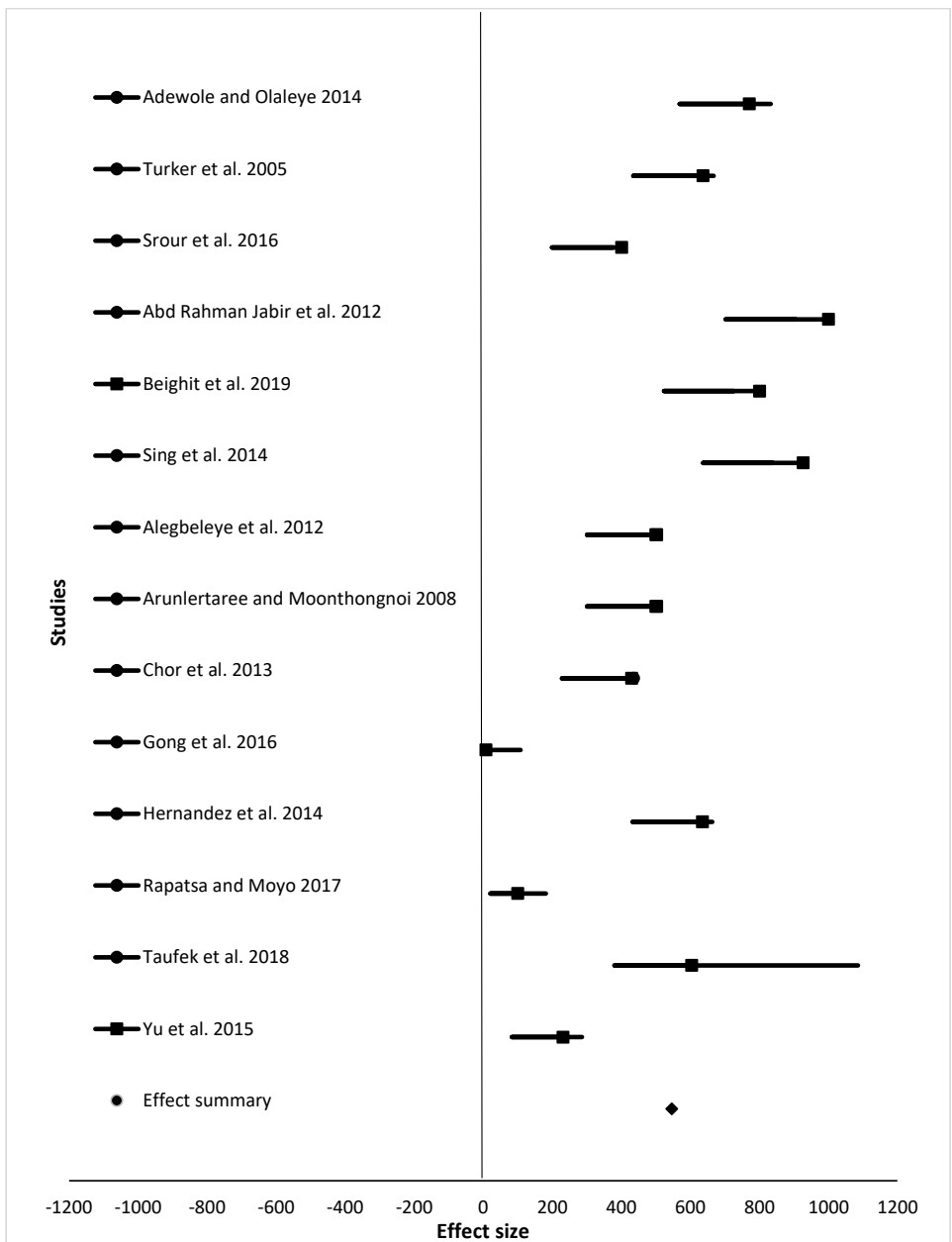

**Figure 5.** The effect size of survival rate (%) of fish from different studies fed different animal protein sources compared to fishmeal as a protein source.

## 4. Discussion

From the results of this review, a variety of fish species, sizes, and inclusion levels have been used in aquaculture (Table 2). A variety of fish species, sizes, and inclusion levels may be because aquaculture is an incredibly diverse industry in terms of species cultured and production systems used [1]. Different fish species have different nutrients requirements [37], which affect the level of protein source inclusion in tested diets. According to [38], human health benefits, competitive price, fish safety, efficiency, customer acceptance, minimal contamination, and ecosystem stress are factors in selecting feeds.

Growth performance measured by final weight and specific growth rate showed that excess protein could not be used efficiently for growth because of growth energy used for the deamination and excretion of absorbed excess amino acids. After all, each fish species had a specific protein limit [39]. According to [40–43], when dietary protein levels increase, the feed conversion ratio decreases. Results from this review indicated that

*O. mossambicus* and *C. gariepinus* fed insect meal (*I. belina*, *Z. variegatus* and *G. bimaculatus*), respectively, converted their feeds efficiently. Both freshwater and marine fish species utilize insects as part of their natural diet [44]. Insects are rich in amino acids, lipids, vitamins, and minerals [45], and they do not require arable land, water, or energy to reproduce [46]. Besides, insects are more natural to replicate, have a higher growth rate, and very effectively transform low-grade or organic matter into high-value protein quite efficiently [44,47].

Recommended levels reported for insect meal in this review shows that a total fishmeal replacement has not been successful. Results support findings reported by [44], who suggested dietary unbalance or deficiencies as the main reason. According to [48], limitations of using insects include their (i) varying nutritional value, which is dependent on the species, stage of development, and the substrate used to feed the insect, (ii) low concentration of sulfur-containing amino acids, and (iii) absence of eicosapentaenoic and docosahexaenoic.

*Oreochromis niloticus*, *C. gariepinus*, *L. guttatus*, *O. niloticus*, and *Op. argus* also efficiently converted terrestrial by-products (fermented feather meal, feather meal, poultry by-products, poultry by-products, and meat and bone meal). Like other animal protein sources, fishery by-products (fish silage, shrimp head meal, and krill meal), fed to Red tilapia (*O. mossambicus* × *O. niloticus* × *O. aureus*), *C. gariepinus* and *A. glueldenstaedtii*, respectively resulted in acceptable feed conversion ratios of 1.35, 2.50, and 1.10, respectively. Survival rates ranged from 83% to 100%, except for Red tilapia (*O. mossambicus* × *O. niloticus* × *O. aureus*) and *C. gariepinus*, which was not reported.

Fermented feather meal, blood meal, poultry by-products, feather meal, meat and bone meal are some of the terrestrial animal by-products used in aquaculture diets [15–19]. Terrestrial by-products have been reported to have great potential as fishmeal replacement because they are readily available, economical sources of protein and have more complete amino acid profiles than vegetable proteins [23]. The use of feather meal in aquaculture feeds is limited by the fact that fish are unable to digest it. Lysine, methionine, and isoleucine have been reported as limiting essential amino acids in poultry by-products, meat and bone meal, and blood meal, respectively [22]. Consumer acceptance is the primary constraint on the use of rendered animal products [23].

Fishery by-products are products generated from fishery industries [41]. Skin and fins, scales, heads and bones, viscera, and muscle trimmings are the main by-products produced in fishery industries with (1–3%), (5%), (9–15%), (12–18%), and (15–20%), respectively [41].

Scanty information is available for these by-products as a fishmeal replacement in fish feeds as they are considered waste [7]. Limiting factors of using fishery by-products include the cost of the collection of fish waste, timely processing, and quality control [49]. Furthermore, fish waste varies highly in its physical nature and proximate composition; and some fish waste such as from seafood is only available during the fishing season [17].

One of the advantages of meta-analysis is to increase the sample size. Samples analyzed in this study were 1335, 1430, 1450, and 1307 for final weight, specific growth rate, feed conversion ratio, and survival rate, respectively. Sample size differs due to the number of studies (15, 17, 18, and 14 for final weight, specific growth rate, feed conversion ratio, and survival rate, respectively) included in the meta-analysis. Results for final weight, specific growth rate, feed conversion ratio, and survival rate (Figures 2–5), shows that there is a statistically significant difference among studies (the overall effect size of the overall effect size of 9015 (95% confidence interval (CI) 6110058.3 to 6110177.58), 10 (95% CI 32 to 21), 10 (95% CI 24 to 13), and 546 (95% CI 350 to 572) for final weight, specific growth rate, feed conversion ratio, and survival rate, respectively. The level of heterogeneity ($I^2$ index) was very high for both the final weight and survival rate with values 99.98 and 101.08, respectively. There was no heterogeneity for both specific growth rate and feed conversion ratio, as their values for I2 index were $I^2 = -25.79\%$ and $I^2 = -17.73\%$, respectively. Final weight, specific growth rate, feed conversion ratio and survival rate of fish in experiment or in farming in general are affected by many factors such as age of fish,

fish species, stocking density, feeding level and frequency, protein source, and water quality parameters such as water temperature, dissolved oxygen, and pH. As shown in Table 2, variety of fish species, size, inclusion levels, recommended levels of protein found were reported, and these are the reasons our meta-analysis indicated heterogeneity in studies. Despite all the heterogeneity observed, these animal protein sources have shown positive effects on feed conversion ratio, specific growth rate, final weight, and survival of different fish species of different size groups.

## 5. Conclusions

Despite the limitations in the use of insects, terrestrial by-products, and fishery by-products as replacement of fishmeal, these animal protein sources have shown positive effects on feed conversion ratio, specific growth rate, final weight, and survival of different fish species of different size groups. However, future studies have recommended to (i) identify a fishmeal replacement that has no limitations, (ii) assessing the suitability of readily available animal meat or by-products as fishmeal replacement.

**Author Contributions:** Conceptualization, R.L.-R.; S.M. and G.O.; methodology, R.L.-R.; data collection, R.L.-R.; data analysis, R.L.-R.; writing—original draft preparation, R.L.-R.; writing—review and editing, S.M., G.O. and R.L.-R.; supervision, S.M. and G.O. All authors have read and agreed to the published version of the manuscript.

**Funding:** This research was funded by the Agribusiness Development Agency (ADA), Pietermaritzburg, South Africa as part of PhD studies for R.L.R, grant number CLI/03.

**Acknowledgments:** We would like to thank the Agribusiness Development Agency (ADA), Pietermaritzburg, South Africa for financial support.

**Conflicts of Interest:** The authors declare no conflict of interest. The funders had no role in the design of the study; in the collection, analyses, or interpretation of data; in the writing of the manuscript, or in the decision to publish the results.

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
