# Peer review of "Animal Protein Sources as a Substitute for Fishmeal in Aquaculture Diets: A Systematic Review and Meta-Analysis"

_applsci, doi:10.3390/app11093854_

Round 1

Reviewer 1 Report

This MS is a review which selects certain published paper for a meta-analysis on papers searching animal-based feed resources as alternatives for fish meal in aquafeeds. The topic is of interest but in my point of view, the systematic review does not include enough published paper into the meta-analysis to really conclude on the set general objective. Apart from this, sometimes the sentences are written in a way which are not correct from an aquaculture nutritionist point of view.

Objective: The objective is very general, which is suitable if much more paper would have been taken into consideration within the meta-analysis. From the 780 selected papers, only 11 publications were included into the meta-analysis and the papers differed in the used fish species and size classes. In short, the inclusion of 11 papers on different species and size classes is to little data to conclude about potential animal-based feed sources to replace fishmeal.

Selection of search terms: Please check the applied search terms. There might be other meaningful search terms (or other technical terms used in aquaculture research like "animal-based" or "resource" instead of "source") to screen for suitable publications for this objective which you otherwise might miss with the selected terms. Further, please check the used combination of terms (esp. AND). I think you might find also additional useful papers if you search for "fishmeal replacement" or "alternative feed ...." alone instead of combining it with another terms. However, with these selected terms you might have missed alternative feed resources (like worms, blood meals).

Selection of papers used: In line 91 you state that you were excluding papers which "did not follow the requirements" without stating the requirements. If the only requirements were the needed data on SGR, FCR, final body mass and mortality, then I would expect much more paper to be available then 11 papers (e.g. in Fig 1 you state one of the requirements to be the inclusion of a measure of data variation (here "standard error")). So, please name all the requirements used to select the papers.

Species included: Please describe the selection of fish species included into the analysis as the used species are not really depicting the important aquaculture species globally (which is fine but needs to be described).

Abstract: I would exclude the section about Nile crocodile from the abstract as this was not matter of your analysis and also in the entire MS this resource is just mentioned in the conclusion.

Introduction: please reduce the information about fish oil from the introduction as this is not matter of your objective. Please also differentiate between expected future needs for food and feed. 

M&M: please describe the used method and procedure of your meta-analysis more in detail.

Table 2: Please describe the table 2 more in detail. To me the given units in combination to the numbers are misleading for the readers.

Figure 2-5: Please describe the figures more in detail.

Conclusion: Please exclude the section of Nile crocodile as it was not named in the entire MS.

Author Response

Dear Reviewer 1,

Kind regards,

Rendani

Reviewer 2 Report

In this paper, the authors selected eleven studies that performed replacement of fishmeal fed to fish with other animal protein source, and described the diversity among the studies.

This paper would be use for providing an overview of recent studies on fishmeal alternatives.
Nevertheless, the statistical data shown at Results are not discussed enough to be systematic review. A quantitative consideration of the variation in values of effect size between studies (Figures 2-5) should be described. Discussion needs major correction or addition to clarify the significance of meta-analysis.

Additionaly, a part that needs minor correction was found in Figure 2. The position of the square indicating the value of study No. [15] should be modified.

Author Response

Dear Reviewer ,

Kind regards,

Rendani

Round 2

Reviewer 2 Report

The manuscript has been well improved by increasing of the number of articles to be analyzed.

I think this revised version is acceptable for publication.